# The Composition of Small Extracellular Vesicles (sEVs) in the Blood Plasma of Colorectal Cancer Patients Reflects the Presence of Metabolic Syndrome and Correlates with Angiogenesis and the Effectiveness of Thermoradiation Therapy

**DOI:** 10.3390/jpm13040684

**Published:** 2023-04-19

**Authors:** Natalia V. Yunusova, Dmitry A. Svarovsky, Artem I. Konovalov, Dmitry N. Kostromitsky, Zhanna A. Startseva, Olga V. Cheremisina, Sergey G. Afanas’ev, Irina V. Kondakova, Alina E. Grigor’eva, Sergey V. Vtorushin, Elena E. Sereda, Anna V. Usova, Svetlana N. Tamkovich

**Affiliations:** 1Department of Biochemistry and Molecular Biology, Central Research Laboratory, Siberian State Medical University, 634050 Tomsk, Russia; bochkarevanv@oncology.tomsk.ru (N.V.Y.); svarovsky.d.a@gmail.com (D.A.S.); oncology-group@yandex.ru (A.I.K.); 2Cancer Research Institute, Tomsk National Research Medical Center, Russian Academy of Sciences, 634009 Tomsk, Russia; d.n.kostromitsky@tomonco.ru (D.N.K.); zhanna.alex@rambler.ru (Z.A.S.); kondakova@oncology.tomsk.ru (I.V.K.);; 3Institute of Chemical Biology and Fundamental Medicine, Siberian Branch of Russian Academy of Sciences, 630090 Novosibirsk, Russia; 4V. Zelman Institute for Medicine and Psychology, Novosibirsk State University, 630090 Novosibirsk, Russia

**Keywords:** matrix metalloproteinases, heat shock proteins, adipose tissue, extracellular vesicles, FABP4, colorectal cancer, thermoradiotherapy efficiency

## Abstract

The majority of colorectal cancer patients (CRCPs) develop tumors on the background of “metabolically healthy obesity” or metabolic syndrome. The aim of the work was to study the levels of matrix metalloproteinases (MMPs) and heat shock proteins (HSPs) on the surface of blood plasma CD9-positive and FABP4-positive small extracellular vesicles (sEVs) from CRCPs depending on metabolic status and tumor angiogenesis, as well as to evaluate the sEVs markers as predictors of the effectiveness of thermoradiotherapy. In CRCPs, compared with patients with colorectal polyps (CPPs), the proportion of triple positive EVs and EVs with the MMP9+MMP2-TIMP1+ phenotype increased significantly among FABP4-positive EVs (adipocyte-derived EVs), which in general may indicate the overexpression of MMP9 and TIMP1 by adipocytes or adipose tissue macrophages in CRCPs. The results obtained have prospects for use as markers to clarify cancer risk in CPPs. One can assume that for CRCPs with metabolic syndrome or metabolically healthy obesity, it is the FABP4+MMP9+MMP2-TIMP1- population of circulating sEVs that is the most optimal biomarker reflecting tumor angiogenesis. Determining this population in the blood will be useful in monitoring patients after treatment for the early detection of tumor progression. CD9+MMP9+MMP2-TIMP1- and MMP9+MMP2-TIMP1+ subpopulations of circulating sEVs are the most promising predictors of the efficacy of thermoradiation therapy because their levels at baseline differ significantly in CRCPs with different tumor responses.

## 1. Introduction

It is known that in the majority of patients with colorectal cancer (CRC), the tumor develops on the background of “metabolically healthy obesity” or metabolic syndrome (more than 60% of patients) [1,2]. Metabolic changes are also characteristic of patients with colon polyps (CPs), with polyps being considered by oncologists to be a precancerous disease. Adipocytes are known to produce the matrix metalloproteinases (MMPs) MMP-2 and MMP-9 and their inhibitor, TIMP1. MMPs derived from adipose tissue play an important role in adipogenesis, angiogenesis, and the remodeling of the extracellular matrix, which can accelerate the development of cancer [3].

Small extracellular vesicles (sEVs) are known to be rich in tetraspanins, with CD9, CD63, and CD81 being vesicular markers. Other tetraspanins important for sEV biogenesis are Tspan8 and CD151 [4]. The main functions of CD151 are the maintenance of epithelial cell integrity, platelet aggregation, regulation of membrane fusion, cell motility, and involvement in angiogenesis and tumor metastasis. CD151 is intracellular in endosomal and lysosomal vesicles, so it can be released from cells as part of sEVs (exosomes) [5]. Tspan8 can form tetraspanin networks with various membrane proteins of sEVs, including CD151. In CRC, Tspan8 has proinvasive potential by interacting with α6β4-integrin, protein kinase C, E-cadherin, claudin-7, EpCAM, and CD44 [6].

Because adipose tissue has a complex cellular composition, sEVs with different functions can be secreted by both adipocytes and stromal cells [7]. In addition, macrophages are found in adipose tissue, accounting for almost 50% of the total number of adipose tissue cells. Various markers for vesicles of adipocytic origin (PPAR gamma, FABP-4, and PREF-1) are discussed in the literature, but none of them are strictly specific to adipose tissue cells, and they can be secreted by some immune (often macrophages) and endothelial cells [8,9,10]. Moreover, adiponectin in exosomes is only a small fraction of the total amount of adiponectin secreted by 3T3-L1 cells and therefore cannot be a reliable marker of plasma sEVs [11]. Thus, FABP4-positive sEVs circulating in the blood may be of both adipocytic and macrophage origin. It was shown that fractions of sEVs of adipocytic origin are specifically enriched in extracellular matrix proteins, including MMPs, chaperones, and some metabolic enzymes involved in lipid and carbohydrate synthesis [12].

Overexpression of MMP2/MMP9 after incubation of recipient tumor cells with sEVs from irradiated donor cells is considered one of the main mechanisms of the non-target effects of radiotherapy [13,14]. The role of MMPs, their inducers, and tissue inhibitors in the processes of invasion, metastasis, and the epithelial–mesenchymal transition is well known, but the role of MMPs and their regulators in sEVs as predictors of radiotherapy effectiveness has not been studied. The role of extracellular and vesicular forms of heat shock proteins (HSPs), which are members of the highly conserved HSP family (HSP90, HSP70, HSP60, HSP27, crystalline, etc.), has been discussed in the mechanisms of radiotherapy and hyperthermia efficacy [15,16]. Since HSPs on the cell surface and sEVs can be recognized by CD91+ tumor cells, CD91+ fibroblasts, CD91+SREC1+TLR+ antigen-presenting cells, and CD94+ cytolytic immune cells, both immunostimulatory and immunosuppressive roles of sEV HSPs have been discussed [17].

The aim of this work was to study the levels of MMPs and HSPs on the surfaces of CD9- and FABP4-positive plasma sEVs in CRC patients (CRCPs) in relation to metabolic status with markers of angiogenesis in the primary tumor, as well as to evaluate markers in sEVs as predictors of the effectiveness of thermoradiotherapy.

## 2. Materials and Methods

### 2.1. Patients and Treatment

The study included 22 CRCPs (10 men and 12 women, T2-4N0-2M0, mean age 59.6 ± 1.6 years) who were treated at the Department of Abdominal Oncology of the Cancer Research Institute of the Tomsk National Research Medical Center from 2019 to 2021. The comparison group included 10 patients (6 men and 4 women, mean age 52.1 ± 2.3 years) examined in the endoscopy department of the Cancer Research Institute of the Tomsk National Research Medical Center, in whom colorectal malignancy was excluded by videocolonoscopy. Adenomatous polyps of the colon were detected in these patients. The exclusion criteria for forming the CRC group were primary multiple forms of CRC and stage Ia cancer (T1N0M0), as well as middle and low rectal cancer. All CPPs and CRCPs had metabolic syndrome in accordance with the IDF (2005) criteria, or so-called “metabolically healthy obesity”, which meant abdominal obesity according to the IDF (2005) criteria, a body mass index ≥30 combined with one of the additional criteria for metabolic syndrome, or isolated obesity [18].

In addition, a group of patients with middle and low rectal cancer (RC) (8 men and 6 women, T3-4N0-1M0, mean age 57.6 ± 1.8 years) was formed to study the possibility of using MMP and HSP expression on blood plasma sEVs as predictors of thermoradiotherapy. Magnetic resonance imaging (MRI) of the pelvis and endoscopic ultrasonography (EUS) of the rectum were used for RC staging. All patients with RC received remote gamma therapy on a Theratron Equinox 1.25 MeV multifractionation device (Best Theratronics, Ottawa, ON, Canada) (1.3 Gy, 2 fractions/day, 5 days/week for 4 weeks, up to a total dose of 54 Gy) and simultaneously received chemotherapy with capecitabine (825 mg/m^2^, twice daily at 12-h intervals) combined with local hyperthermia using a Celsius TCS machine (Celsius 42 GmbH, Eschweiler, Germany) (3 times per week, 3 h before the radiation therapy session at 42–44 °C for 45–60 min, 10 sessions total). Treatment efficacy was assessed 6–7 weeks after the completion of radiation therapy using response assessment criteria RECIST 1.1. and ESGAR [19].

All patients underwent MRI on a MAGNETOM ESSENZA 1.5 Tesla MRI scanner (Siemens, Erlangen, Germany) with a Body Matrix surface phased coil. The MRI protocol included T2-weighted images in the axial, sagittal, and coronal planes (TR-3310 ms, TE-99 ms, slice thickness-3.5 mm, FA-150°, FOV-380 × 309, Matrix-512 × 384), T1-weighted images in the axial plane (TR-861 ms, TE-12 ms, slice thickness-3. 5 mm, FA-150°, FOV-298 × 268, matrix-320 × 240), and DWI (TR-2600 ms, TE-92 ms, slice thickness-3.5 mm, FOV-250 × 250, matrix-128 × 128, b values 50–400–800 s/mm^2^) with automatic construction of parametric ADF maps. Software version Singo MR C15 (Siemens, Erlangen, Germany) was used. Two hours before the study, patients took 2 capsules of buscopan. To assess tumor response to thermoradiotherapy we used the mrTRG system, visual DWI, and ADS analysis.

All cancer patients were evaluated for response to treatment and were divided into two groups: those with a complete response and those with a partial response or stable disease. Patients with complete response had no measurable tumors, and the tumor bed showed fibrosis with intermediate signal intensity on T2-VI (mrTRG 1) without a high signal on DWI and a low signal on ADS. Patients with partial response or stable disease had residual tumors, areas of fibrosis with intermediate signal intensity on T2-weighted images (mrTRG -2–4), and areas with high signal on DWI and low signal on ADS.

Patients with pathological complete responses were under observation. Patients with partial tumor regression were operated on 7–8 weeks after completion of chemoradiation therapy.

### 2.2. Measurement of Anthropometric Parameters and Blood Plasma Metabolic Markers

Waist circumference was measured as the circumference in centimeters halfway between the lower rib and the iliac crest. Hip circumference was measured at the point yielding the maximum circumference over the buttocks. Body mass index was calculated as the weight in kilograms divided by the square of the height in meters. The study of the plasma level of glucose, total cholesterol, HDL cholesterol, LDL cholesterol, and triglycerides was carried out on a multi-channel biochemical analyzer Konelab-20 (Thermofisher Scientific, Vantaa, Finland), after a 16-h fast using reagents from Thermofisher Scientific (Vantaa, Finland).

### 2.3. Immunohistochemical Staining and Assessment

Tumor tissue samples were examined using an automated system for immunohistochemical staining and in situ hybridization Bond™-MAX (Leica Biosystems, Deep Park, TX, USA). Tumor tissues were stained with antibodies to CD31 (clone JC70A. RTU, Dako, Glostrup, Denmark) and VEGFA (Clone EP1176Y, dilution 1:100, Abcam, London, UK).

Using light microscopy (Nikon Eclipse Ni), we counted microvessels in the 200× field in the most active areas of neovascularization and measured microvessel density. Vessels stained with anti-CD31 monoclonal antibodies had three main morphological features: rounded, sinus-like, and small vessels without distinguishable lumens. When calculating microvessel density, only vessels with distinguishable small-caliber lumen without thick muscular walls were considered. The procedure used corresponds to the international consensus method of intratumoral microvascular density (MVD) assessment [20]. The immunostained slice was evaluated at low magnification (×40), and the three areas with the greatest number of well-defined microvessels (so-called hot spots) were chosen. Microvessels were counted at ×200 magnification. Microvessel density results were expressed as the average number of vessels per field of high magnification (×200) in the three selected hot spots. VEGFA expression was assessed by the cytoplasmic staining intensity in tumor stromal elements (1—negative, 2—weak, 3—moderate, 4—strong).

### 2.4. Small EVs Isolation, Electron Microscopy, and Nanoparticle Tracking Analysis

Blood plasma small EVs (sEVs) were isolated using ultrafiltration with ultracentrifugation as previously described [21]. Briefly, venous blood (18 mL) was collected in K3EDTA spray-coated vacutainers, placed at +4 °C, and processed within an hour after taking the blood. The blood cells were pelleted by centrifugation for 20 min at 1000× *g* and 4 °C (bucket rotor, Labofuge 400R, Thermo Fisher, Bremen, Germany). To remove the cell debris, plasma samples were centrifuged for 20 min at 17,000× *g* and 4 °C (angular rotor, centrifuge 5415R, Eppendorf, Hamburg, Germany). To remove microvesicles and apoptotic bodies, the supernatant was diluted 4–5-fold with PBS (10 mM phosphate buffer, 0.15 M NaCl, pH 7.5) and passed through a 100 nm pore-size filter (Minisart high flow, 16553-K, Sartorius, Goettingen, Germany). Then, the filtrate was centrifuged at 100,000× *g* (bucket rotor, Optima XPN 80, Beckman Coulter, Brea, CA, USA) and 4 °C for 90 min. The pellet was resuspended in 10 mL PBS and centrifuged twice under the same conditions. Then supernatants were removed, and the pellets were resuspended in 300 µL PBS. sEV samples were aliquoted and stored at −80 °C. The aliquots were thawed once before use.

Blood sampling for obtaining sEVs in patients was performed at the stages of combined treatment at four points: the 1st point—before treatment; the 2nd point—the middle of the course of thermoradiotherapy with the radiomodifier capecitabine; the 3rd point—6–7 weeks after the end of thermoradiotherapy (about 3 months after the first point); the 4th point—6 months after the first point. The interval of 3 months for taking the checkpoint was reasonable due to the fact that during this period, the effect of the course of thermoradiotherapy was fully realized (the maximum indicators of the immediate effectiveness of treatment were reached). In addition, in accordance with the requirements of clinical guidelines, it was necessary to observe the following frequency of control examination of patients after the completion of CRC treatment in order to diagnose tumor progression in a timely manner: in the first 1–2 years, the examination is recommended every 3–6 months.

For negative staining, 10 μL of isolated sEVs was adsorbed for 1 min on copper grids covered with carbonized formvar film. Then the grids were exposed for 5–10 s on a drop of 0.5% uranyl acetate. Grids were studied using Jem 1400 transmission electron microscope (Jeol, Tokio, Japan), and the images were obtained with a digital camera Veleta (EM SIS, Muenster, Germany).

The size distribution and concentration of the sEVs were measured by nanoparticle tracking analysis (NTA) using the NanoSight^®^ LM10 (Malvern Instruments, Malvern, Worcestershire, UK) analyzer equipped with a blue laser (45 mW at 488 nm) and a C11440-5B camera (Malvern Instruments, Malvern, Worcestershire, UK). To optimize the measurement mode, the samples of sEVs were diluted 1:100, 1:1000, or 1:10,000 by PBS. In the selected dilution, each sample was measured in triplicate. Recording and data analysis were performed using NTA software 2.3 (Malvern Instruments, Malvern, Worcestershire, UK). The following parameters were evaluated during the analysis of the recording monitored for 60 s: the average hydrodynamic diameter, the mode of distribution, the standard deviation, and the concentration of vesicles.

### 2.5. Flow Cytometry

To evaluate protein concentrations, 7.5 μL of an sEV sample was mixed with 2.5 μL of lysis buffer (0.25 M Tris-HCl, 8% SDS, 0.2 M DTT, pH 6.8), incubated on ice (10 min), boiled (95 °C for 10 min), and cooled. The protein concentration was measured using a fluorometric protein assay (NanoOrange^®^ Protein Quantitation Kit, Molecular Probes, Eugene, OR, USA). The fluorescence of the samples was measured with a Cytation™ 1 Cell Imaging Multi-Mode Reader (Agilent Tech., Santa Clara, CA, USA).

The 4 μm-diameter aldehyde/sulfate latex beads (Molecular Probes, Eugene, OR, USA) in MES buffer were incubated with anti-CD9 (ab134375, London, Abcam) antibodies at room temperature for 14 h with gentle agitation. The aliquots of sEVs (about 30 μg vesicular protein) were incubated with 3 × 10^5^ antibody-coated latex beads in 150 μL of PBS at 4 °C for 14 h with gentle agitation. The reaction was blocked with 0.2 M glycine for 30 min at 4 °C. The sEV–antibody–bead complexes were washed twice with washing buffer (2% EVs depleted bovine serum in PBS) and incubated with a blocking immunoglobulin G (BD Biosciences, Heidelberg, Germany) at room temperature for 10 min with washing.

#### 2.5.1. Analysis of CD9/CD63/CD81/CD24 Subpopulations in Plasma sEVs

Incubation with FITC-conjugated antibodies against tetraspanins (CD63, CD81) and antibodies against CD24 (BD Biosciences, Heidelberg, Kennesaw, GA, USA) was performed at 4 °C for 50 min.

The complexes were washed twice with washing buffer. Flow cytometry was performed on a cytometer Cytoflex (Becman Coulter, BioBay, Shanghai, China). Data were analyzed with CytExpert 2 software (Becman Coulter, Pasadena, CA, USA). The median fluorescence intensity (MFI) of the EVs was analyzed in comparison with the isotypic and negative controls (BD bioscience, Louisville, CO, USA).

#### 2.5.2. Analysis of HSP60/HSP27/HSP90 Subpopulations on the Surface of CD9-Positive sEVs

After washing the CD9 antibody-coated latex bead–sEV complexes with PBS, human BD Fc Block (564219, BD, San Jose, CA, USA) was used to block nonspecific binding, and then the complexes were stained with antibodies (anti-HSP60-PE (2 µL per test, FAA822Hu41, Cloud-Clone Corp., Wuhan, China), anti-HSP27-FITC (2 µL per test, FAA693Hu81, Cloud-Clone Corp., Wuhan, China), and anti-HSP90-APCF (2 µL per test, FAA863Hu51, Cloud-Clone Corp., Wuhan, China) for 20 min at room temperature. Individual complexes were gated and examined on a Cytoflex cytometer. 

#### 2.5.3. Analysis of MMP9/MMP2/TIMP1 Subpopulations on the Surface of CD9-Positive sEVs

After washing the CD9 antibody-coated latex bead–sEV complexes with PBS, human BD Fc Block (BD Biosciences, Heidelberg, Germany) was used to block nonspecific binding, and then the complexes were stained with anti-TIMP1-APC (2 µL per test, FAA522Hu51, Cloud-Clone Corp., Wuhan, China), anti-MMP2-PE (2 µL per test, FAA100Hu41, Cloud-Clone Corp., Wuhan, China), and anti-MMP9- FITC antibodies (2 µL per test, FAA553Hu81, Cloud-Clone Corp., Wuhan, China). Individual complexes were gated and examined on a Cytoflex cytometer. 

#### 2.5.4. Analysis of HSP60/HSP27/HSP90 and MMP9/MMP2/TIMP1 Subpopulations on the Surface of FABP4-Positive sEVs

Analysis of HSP60/HSP27/HSP90 and MMP9/MMP2/TIMP1 Subpopulations on the Surface of FABP4-Positive sEVs was performed similarly to the above method. Monoclonal anti-FABP4-antibody (ab134375, Abcam, London, UK) was used.

#### 2.5.5. Analysis of Tspan8/CD151 Subpopulations in Plasma EVs. Evaluation of the Tspan8/CD151 Tetraspanin Subpopulations in Plasma sEVs

Analysis of Tspan8/CD151 Subpopulations in Plasma EVs. Evaluation of the Tspan8/CD151 tetraspanin subpopulations in plasma sEVs was performed similarly to the method described above. Anti-Tspan8-PE antibody (3 μL per test, ABIN4895321Antibodies-online, Germany) and anti-CD151-APC antibody (3 μL per test, 350405 Biolegend, San Diego, CA, USA) were used.

All flow cytometry investigations were carried out at the Core Faculty “Medical Genomics”, Tomsk National Research Medical Center (Tomsk).

### 2.6. Statistical Analysis

Statistical analysis was carried out using Statistica 10 (TIBCO Software, Palo Alto, CA, USA) software. All data were expressed as means with standard errors. Mann–Whitney or Kruskal–Wallis tests were used to evaluate statistical differences between groups, and *p*-values < 0.05 were considered statistically significant. Correlation analysis of the data was carried out with the Spearman Rank Correlation test. *p*-Values < 0.05 were considered statistically significant. Differences in the HSP and MMP composition of sEVs for RCPs with partial and complete tumor response were assessed using the repeated measures ANOVA test.

## 3. Results

### 3.1. Characteristics of Isolated sEVs

Transmission electron microscopy, NTA, and flow cytometry were used to characterize sEV isolated from the blood plasma.

Transmission electron microscopy revealed clearly structured, low electron density cup-shaped particles with preserved membranes in a preparation of sEVs isolated from the blood plasma of CPPs and CRCPs (Figure 1A). It was found that their morphology did not differ from that of particles isolated from patients with other types of cancer [22,23].

Particles called “non-vesicles” (low- or very-low-density lipoproteins) were also present in the samples. The lipoproteins could not be separated from the sEV preparations by any of the methods used to isolate sEVs from body fluids.

NTA showed that the sEVs of blood plasma CPPs and CRCPs had similar sizes: the sEVs of plasma CPPs had a mean size of 108 nm with a mode of 71 nm and an SD of 50 nm; the sEVs of plasma CRCPs had a mean size of 94 nm with a mode of 70 nm and an SD of 56 nm (Figure 1B). The mean concentration of EVs isolated from plasma CPPs and CRCPs was not significantly different (39.7 ± 7.0 × 10^9^ particles/mL blood and 30.8 ± 3.9 × 10^9^ particles/mL blood, respectively).

The sEVs absorbed on aldehyde sulfate latex beads using anti-CD9 antibodies were stained with FITC-labeled antibodies to the CD63 and CD81 tetraspanin family receptors, as well as antibodies to the CD24 receptor. Similar plasma sEV subpopulation compositions were found in CPPs and CRCPs, and a statistically significant difference between polyp and cancer patients was found in the MFI of the CD9/CD24 sEV subpopulation (Figure 1C).

### 3.2. Subpopulations of MMPs and HSPs on the Surface of CD9-Positive and FABP4-Positive Plasma sEVs from CPPs and CRCPs

The gating strategy as well as the composition of MMPs and HSPs on the surface of CD9-positive sEVs in CPPs and CRCPs is shown in Figure 2 and Table 1.

The results presented are consistent with our earlier data on the preferential expression of MMP9 on the surface of CD9-positive sEVs compared with MMP2 and TIMP1 in healthy women, patients with borderline ovarian tumors, and patients with ovarian cancer, and MMP9-positive exosomes were more frequently found in CRCPs [21]. However, triple-positive sEVs expressing MMP9, MMP2, and TIMP1 inhibitor were more frequently found in CPPs. HSP60 was most frequently expressed on the surface of sEVs, and HSP60-positive sEVs were found in CPPs much more frequently than in CRCPs. However, plasma sEVs expressing both HSP60 and HSP27 were found fourfold more frequently in CRCPs (Figure 2A–E).

The composition of MMPs and HSPs on the surface of FABP4-positive sEVs in CPPs and CRCPs with metabolic disorders is shown in Table 2. When analyzing the expression of MMPs, TIMP1, and HSPs on the surface of adipocyte-derived sEVs, no statistically significant differences were found in HSP levels between the groups. The proportion of FABP4-positive sEVs expressing MMP9, MMP2, and their inhibitor TIMP1 was significantly higher in CRCPs compared with CPPs (*p* < 0.05). Similarly, FABP4-positive sEVs expressing MMP9 and TIMP1 but not expressing MMP2 were more frequently found in CRCPs. The results indicate that HSP90 expression on CD9-positive and FABP4-positive EVs in plasma CRCPs and CPPs is not typical (Figure 2A,F–H).

### 3.3. The Expression of MMPs, TIMP1, and HSPs on the Surface of CD9-Positive and FABP4-Positive sEVs Related to Body Mass Index, Plasma Triglycerides, and Plasma HDL Cholesterol Levels in CRCPs

The Spearman correlation coefficients (r) between the phenotypes of CD9-positive, FABP4-positive sEVs and age, body mass index (BMI), waist circumference, triglycerides (TGs), total cholesterol, and HDL-cholesterol levels in CRCP are presented in Table 3 and Table 4, respectively. The correlation analysis performed revealed no statistically significant correlations in the CPPs for both CD9-positive and FABP4-positive sEVs.

Correlation analysis revealed multiple correlations of CD9-positive sEV phenotypes in CRCPs with BMI and plasma HDL-cholesterol levels, while the phenotypes of FABP4-positive sEVs were mainly related to plasma triglyceride levels.

### 3.4. Intratumoral CD31 Expression in CRCPs Related to the Level of FABP4+MMP9+MMP2+TIMP1- and FABP4+MMP9+MMP2-TIMP1- Blood Plasma sEVs

Intratumor VEGF-A and PECAM-1 (platelet/endothelial cell adhesion molecule 1) and CD31 expression in CRCPs were analyzed with the levels of MMPs and their inhibitors as well as the proangiogenic tetraspanins Tspan8 and CD151 on isolated plasma sEVs. Strong positive correlations were found between CD31 levels in primary tumor tissue and FABP4+MMP9+MMP2+TIMP1- (r = 0.82, *p* < 0.05) and FABP4+MMP9+MMP2-TIMP1- (r = 0.67, *p* < 0.05) populations of plasma sEVs (Figure 3A). Examples illustrating the relationship between MMP levels on sEVs and markers of angiogenesis CD31 in the primary tumor are shown in Figure 3B,C. There was no correlation between markers of angiogenesis in primary tumors and the levels of other vesicular proteins examined.

### 3.5. Composition of HSPs and MMPs on the Surface of CD9-Positive sEVs Derived from RCPs during Combined Treatment with Thermoradiotherapy: Relationship with Treatment Efficacy

A single-factor analysis of variance for repeated measures was performed to assess differences in the composition of circulating CD9-positive sEVs between subgroups of patients with complete and partial regression during thermoradiotherapy. Of the 14 patients, 6 patients showed complete tumor regression, and 8 patients showed partial tumor regression or stabilization (see the criteria for complete and partial tumor response in the “Patients and Treatment” section). The data are shown in Figure 4.

The analysis showed that the content of HSP60+, MMP9+MMP2+TIMP-, MMP9+2-TIMP-, and MMP9+2-TIMP+ populations of sEVs differed in patients with complete and partial tumor response. In patients with complete response, the level of HSP60+ sEVs was lower at all checkpoints than in patients with partial regression. It should be noted that the levels of MMP9+MMP2-TIMP1- and MMP9+MMP2-TIMP1+ sEVs in patients with complete and partial response differed significantly even before treatment, which makes it promising to determine these populations as predictors of therapy efficacy. The obtained data indicate prospects for further study of the composition of plasma sEVs for the study of EV-associated mechanisms of thermoradiotherapy efficacy and as accessible and informative markers—predictors of the efficacy of this type of treatment in patients with rectal cancer.

## 4. Discussion

In accordance with our previously published results on the analysis of subpopulations of sEV surface proteins in norm and in various diseases, the expression of CD9 was significantly higher than that of other markers [22,23]. Accordingly, the study of the CD9-positive sEVs as the core population is most appropriate. FABP4 is a popular and frequently used marker of sEVs produced by adipocytes that has certain advantages over other candidate markers (adiponectin, PPAR-Y), such as its high concentration on the surface of sEVs as well as the known mechanism of marker secretion within sEVs [24,25]. The fact that FABP4 is localized on the surface of EVs in human plasma and can be successfully used to visualize EVs of adipocytic origin and EVs of adipose tissue macrophages by flow cytometry was evidenced by the study of Gustafson CM et al. [26]. Our results showed that the levels of surface MMPs and TIMP1 significantly differed on CD9-positive and FABP4-positive sEVs. In particular, the proportion of triple-positive sEVs and vesicles with the MMP9+MMP2-TIMP1+ phenotype significantly increased on FABP4-positive sEVs in CRCPs compared with CPPs, which in general may indicate the hyperexpression of MMP9 and TIMP1 by adipocytes or adipose tissue macrophages in CRCPs. The results obtained have prospects for use as markers to clarify cancer risk as well as in terms of explaining the efficacy of treatment in obese cancer patients.

It has been shown that the profile of HSPs in papillary thyroid cancer tissue and circulating exosomes are identical; in particular, HSP27 and HSP60 predominate, while HSP90 and HSP70 are weakly expressed [27]. Although Western blotting was used to determine the level of HSPs in exosomes in [27] and flow cytometry was used in our work, we obtained similar data on the distribution of HSPs in sEVs of patients with pre-tumor disease and CRC, indicating the predominance of HSP60 and HSP27 on of both CD9-positive sEVs and FABP4-positive sEVs.

Correlation analysis revealed multiple correlations of phenotypes of CD9-positive sEVs in CRCPs with BMI and plasma HDL-cholesterol levels, while FABP4-positive sEV phenotypes were mainly associated with plasma triglyceride levels. Because adipose tissue plays a key role in the development of insulin resistance, the sEVs secreted by adipose tissue may be a kind of information transmitter in this process [1]. It was previously shown that HOMA-B and HOMA-IR insulin resistance indices were related to the concentration of EVs and insulin cascade proteins in EVs. Specifically, high levels of HOMA-IR were associated with lower levels of the ribosomal protein phosphoS6RP [Ser240/244] and activated insulin receptor (phospho-IR) [28].

Tumor-derived EVs can transport proangiogenic molecules into endothelial cells, promoting their angiogenic activity through the VEGF/VEGF to receptor, Notch, Wingless-type (WNT), and Hypoxia-inducible factor (HIF) signaling pathways [29,30]. Among the commonly known proangiogenic signaling pathways, miRNAs, lnRNAs, circRNAs, and proteins carried by tumor-derived sEVs have recently come to be considered modulators of tumor angiogenesis. To date, more than 35 miRNAs, more than 19 lnRNAs, and 6 circRNAs have been identified in tumor-derived EVs that have a stimulatory effect on mouse HUVECs or aortic endothelial cells [31,32]. The novelty of this study is that there is evidence of a strong association between circulating vesicles derived from adipocytes carrying pro-angiogenic MMPs and not expressing their inhibitor TIMP1 with the level of angiogenesis in the primary tumor. It should also be noted that in this clinical group of patients with metabolic disorders, different populations of CD9-positive sEVs expressing the MMP complexes, their inhibitors, and HSPs did not correlate with CD31 and VEGF-A expression in the primary tumor.

## 5. Conclusions

In CRC, compared with CP, the proportion of triple-positive sEVs and sEVs with the MMP9+MMP2-TIMP1+ phenotype increased significantly among FABP4-positive sEVs, which, in general, may indicate the overexpression of MMP9 and TIMP1 by adipocytes or adipose tissue macrophages in CRCPs. The results obtained have prospects for use as markers for CRC risk assessment in CPPs as well as in terms of explaining the effectiveness of treatment in obese CRCPs. Correlation analysis revealed multiple correlations of individual CD9-positive sEV phenotypes in CRCPs with BMI and serum HDL levels, while FABP4-positive sEV phenotypes were associated mainly with triglyceride levels.

We can assume that for RPC patients with metabolic syndrome or metabolically healthy obesity, the FABP4+MMP9+MMP2-TIMP1- population of plasma sEVs is the most optimal biomarker reflecting tumor angiogenesis. The determination of this population in plasma would be useful in monitoring patients after treatment for the early detection of tumor progression, which would be accompanied by increased neoangiogenesis.

Subpopulations of circulating CD9-positive sEVs with the MMP9+MMP2-TIMP1- and the MMP9+MMP2-TIMP1+ phenotypes are the most promising predictors of the efficacy of thermoradiotherapy, as their levels initially differ significantly in RC patients with different responses to thermoradiotherapy.

## Figures and Tables

**Figure 1 jpm-13-00684-f001:**
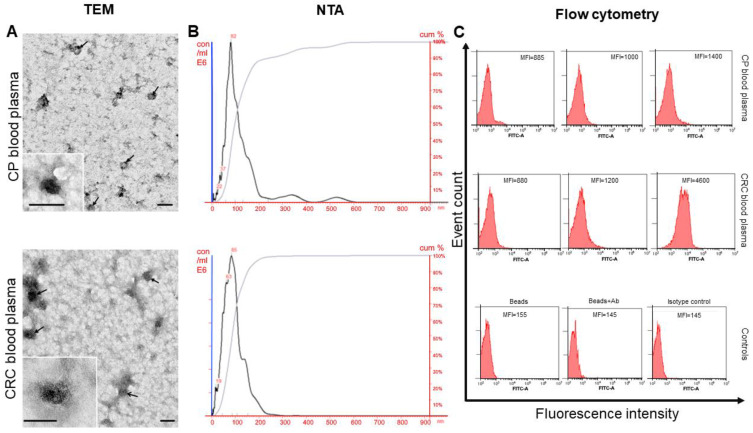
Identification of isolated sEVs. (**A**) TEM showed the presence of vesicles with typical morphology and no vesicles larger than 100 nm. The inserts show sEVs. Scale bars correspond to 100 nm. Electron microscopy, negative staining with uranyl acetate; (**B**) size distribution of plasma sEVs isolated from the blood of CPPs and CRCPs. Data of NTA analysis; (**C**) expression of CD63, CD81, and CD24 on CD9-positive sEVs of blood plasma CPPs and CRCPs. Representative median fluorescence intensity (MFI) values are shown for flow cytometry. Each study was performed in triplicate. For isotype controls (histogram at right), labeled CD9 bead–sEV complexes were incubated with mouse FITC IgG1, k Isotype control or mouse FITC IgG2a, k Isotype control. One of the representative isotypic controls is shown. For the negative control, nothing was added to the CD9 antibody-labeled latex particles (histogram at left) or incubated with FITC-labeled antibodies (anti-CD63, anti-CD81, or anti-CD24). One of the representative negative controls is shown (histogram in the center).

**Figure 2 jpm-13-00684-f002:**
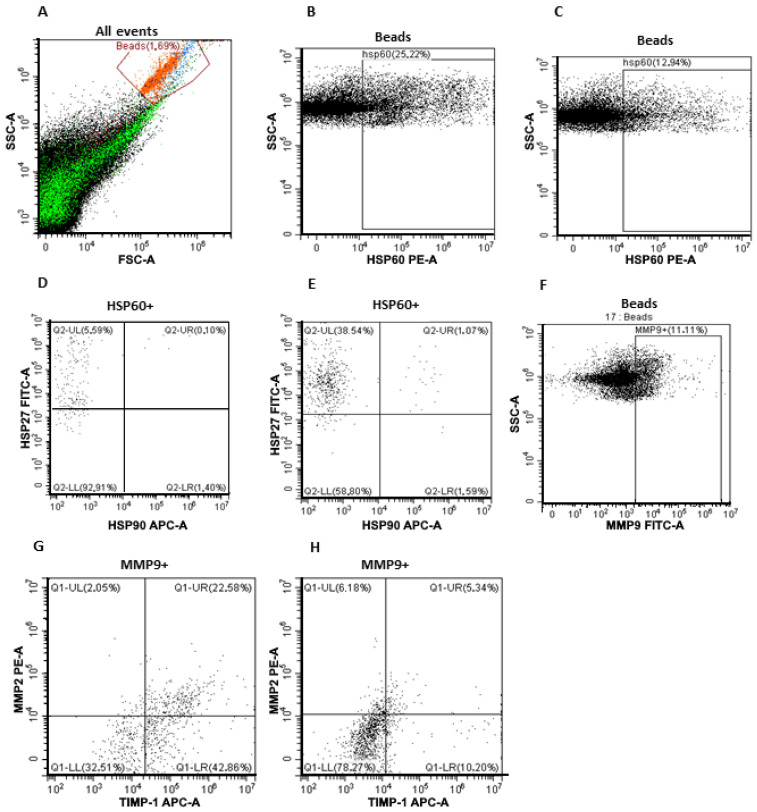
Flow cytometry of blood plasma sEVs. (**A**) Forward scatter area (FSC-A) versus side scatter area (SSC-A) dot plot representing sEV samples adsorbed on aldehyde–sulfate latex beads labeled anti-CD9 or anti-FABP4 antibodies; (**B**) HSP60-positive sEVs within CD9-positive sEVs in CPPs and (**C**) CRCPs; (**D**) triple-labeling with antibodies against HSP60, HSP27, and HSP90 of plasma CD9-positive sEVs of CPPs and (**E**) CRCPs; (**F**) MMP9-positive population within FABP4-positive sEVs in CRCPs; (**G**) triple-labeling with antibodies against MMP9, MMP2, and TIMP1 of blood plasma FABP4-positive sEVs of CRCPs and (**H**) CPPs.

**Figure 3 jpm-13-00684-f003:**
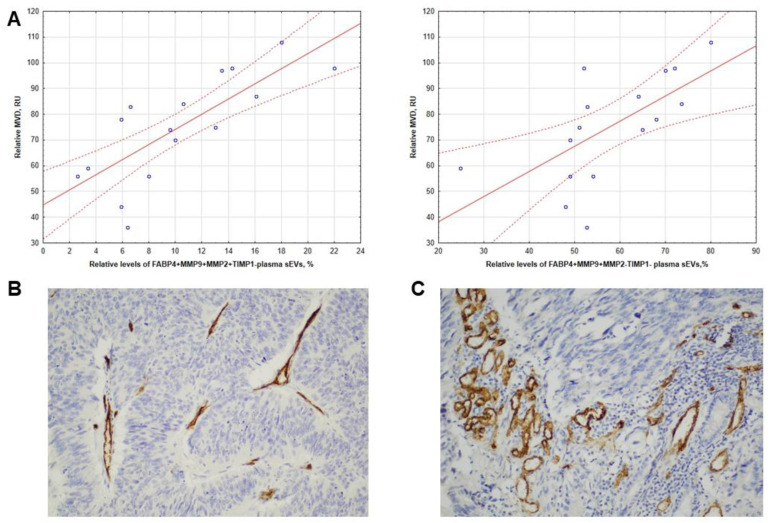
Correlation of FABP4-positive plasma sEVs in CRCPs with intratumoral microvascular density (MVD). (**A**) Correlation of FABP4+MMP9+MMP2+TIMP1- (**left**) and FABP4+MMP9+MMP2-TIMP1- (**right**) plasma sEVs levels with intratumoral MVD; the photographs are from two representative CRCPs; (**B**) case 1 (pT3N0M0, stage II) exhibited low MVD in the stroma of a colorectal adenocarcinoma, and (**C**) case 2 (pT3N2M0, stage III) showed high MVD. immunohistochemistry (CD31). ×200. MDV expressed in relative units (RU) is presented as described above.

**Figure 4 jpm-13-00684-f004:**
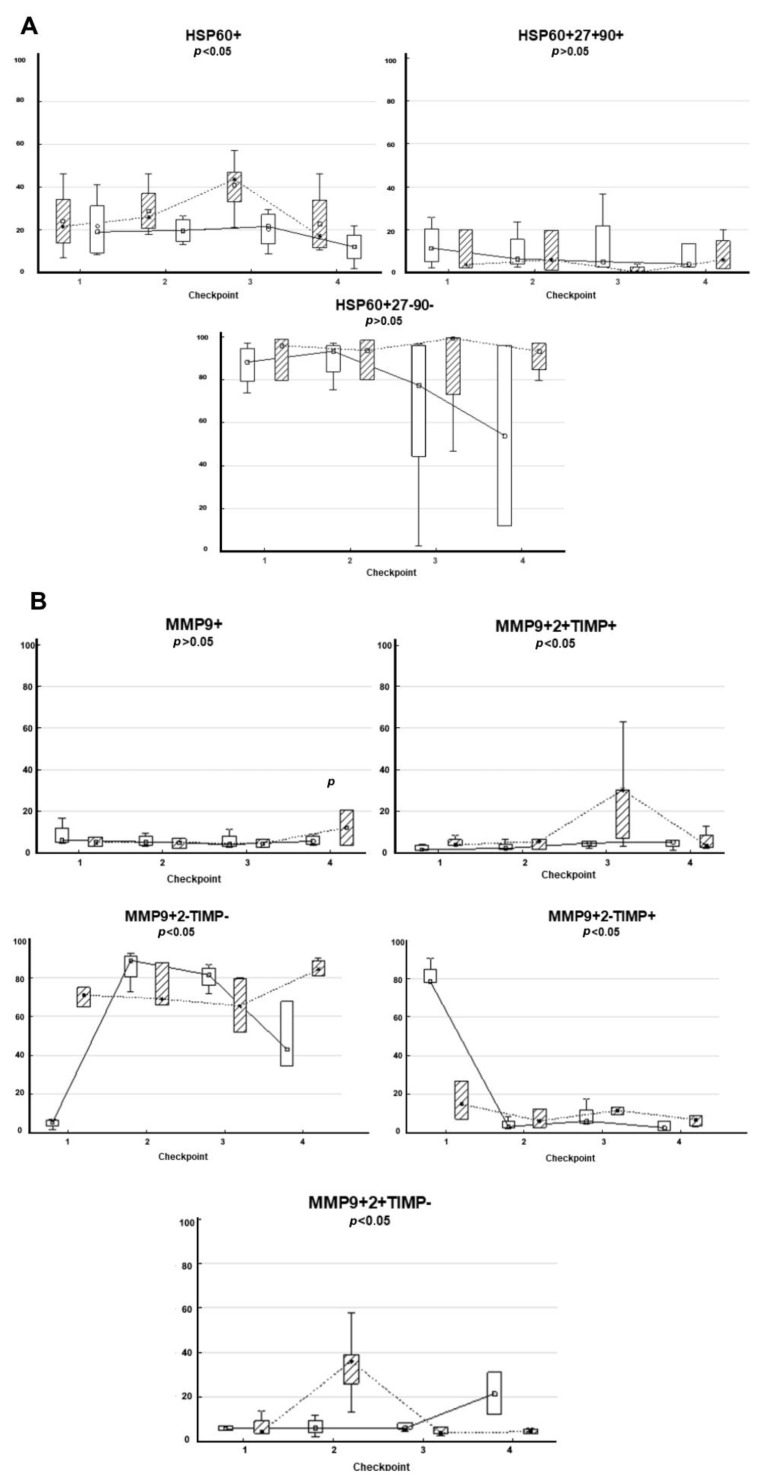
HSPs and MMPs composition at the surface of CD9-positive sEVs of RCPs depending on the effectiveness of thermoradiotherapy. (**A**) HSP and (**B**) MMP composition on CD9-positive sEVs of RCPs depending on the effectiveness of thermoradiotherapy. The dotted line and shaded boxes correspond to partial tumor response and stabilization; the solid line and light boxes are complete tumor responses. The median, 25–75 quartiles, and *p*-level are presented. The x-axis shows checkpoints for monitoring patients during which blood was taken for sEVs isolation. The y-axis is presented as a percentage of the sEVs subpopulation in blood plasma.

**Table 1 jpm-13-00684-t001:** MMPs and HSPs composition on the surface of CD9-positive sEVs in blood plasma in CPPs and CRCPs, mean ± SE *.

Phenotypes of EVs, %	Groups		*p*-Level
	CPPs (*n* = 10)	CRCPs (*n* = 22)	
HSP60+	19.4 ± 1.86	11.10 ± 1.21	**<0.05**
HSP60+HSP27+HSP90-	6.30 ± 3.26	26.5 ± 3.01	**<0.05**
HSP60+HSP27-HSP90-	91.7 ± 3.52	73.0 ± 2.91	**<0.05**
HSP60+HSP27-HSP90+	1.35 ± 0.55	1.49 ± 0.25	
MMP9+	6.45 ± 1.82	11.0 ± 1.02	**<0.05**
MMP9+MMP2+TIMP1+	8.70 ± 1.02	5.80 ± 0.93	**<0.05**
MMP9+MMP2+TIMP1-	11.0 ± 0.92	9.70 ± 1.37	
MMP9+MMP2-TIMP1-	71.7 ± 10.2	75.3 ± 4.50	
MM9+MMP2-TIMP1+	11.4 ± 3.71	10.4 ± 2.85	

* The incidence of individual sEV phenotypes as a percentage of all sEVs.

**Table 2 jpm-13-00684-t002:** MMP and HSP composition on the surface of FABP4-positive sEVs in blood plasma in CPPs and CRCPs, mean ± SE *.

Phenotypes of EVs, %	Groups		*p*-Level
	CPPs (*n* = 10)	CRCPs (*n* = 22)	
HSP60+	18.9 ± 3.10	19.5 ± 2.21	
HSP60+HSP27+HSP90-	24.1 ± 3.90	23.7 ± 3.22	
HSP60+HSP27-HSP90-	74.7 ± 5.55	75.0 ± 5.35	
HSP60+HSP27-HSP90+	0.95 ± 0.35	1.05 ± 0.24	
MMP9+	8.33 ± 1.15	11.2 ± 2.00	
MMP9+MMP2+TIMP1+	3.33 ± 1.71	10.9 ± 2.40	**<0.05**
MMP9+MMP2+TIMP1-	10.6 ± 2.53	10.3 ± 1.77	
MMP9+MMP2-TIMP1-	82.3 ± 5.83	61.3 ± 6.38	**<0.05**
MMP9+MMP2-TIMP1+	3.73 ± 0.65	17.4 ± 3.10	**<0.05**

* The incidence of individual sEV phenotypes as a percentage of all sEVs.

**Table 3 jpm-13-00684-t003:** Spearman correlation coefficients (r) between CD9-positive sEVs phenotypes and age, BMI, waist circumference, TGs, total cholesterol, and plasma high-density lipoprotein (HDL) cholesterol in CRCPs *.

Phenotypes of EVs, %	Age	BMI	Waist Circumference	TGs	Total Cholesterol	HDL Cholesterol
HSP60+						
HSP60+HSP27+HSP90-						0.61
HSP60+HSP27-HSP90-						−0.61
HSP60+HSP27-HSP90+		−0.61				−0.68
MMP9+		0.53	0.60			
MMP9+MMP2+TIMP1+	0.57					
MMP9+MMP2+TIMP1-						
MMP9+MMP2-TIMP1-						
MMP9+MMP2-TIMP1+		0.56				0.62

* The table shows only significant correlation coefficients with a significance level of *p* < 0.05.

**Table 4 jpm-13-00684-t004:** Spearman correlation coefficients (r) between FABP4-positive sEVs phenotypes and age, BMI, waist circumference, TGs, total cholesterol, and HDL cholesterol in CRCPs.

Phenotypes of EVs, %	Age	BMI	Waist Circumference	TGs	Total Cholesterol	HDL Cholesterol
HSP60+						−0.59
HSP60+HSP27+HSP90-						
HSP60+HSP27-HSP90-			0.53	0.54		
HSP60+HSP27-HSP90+		−0.68	−0.79			
MMP9+					−0.74	
MMP9+MMP2+TIMP1+				0.67		
MMP9+MMP2+TIMP1-						
MMP9+MMP2-TIMP1-						
MMP9+MMP2-TIMP1+				0.52		

* The table shows only significant correlation coefficients with a significance level of *p* < 0.05.

## Data Availability

The data presented in this study are available on request fron the corresponding author.

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
