# Peer review of "The Composition of Small Extracellular Vesicles (sEVs) in the Blood Plasma of Colorectal Cancer Patients Reflects the Presence of Metabolic Syndrome and Correlates with Angiogenesis and the Effectiveness of Thermoradiation Therapy"

_jpm, 2023, doi:10.3390/jpm13040684_

Round 1

Reviewer 1 Report

The manuscript is well structured and is based on research from the scientific literature. The authors highlighted the aims, significance and novelty of their study. The paper studies the levels of matrix metalloproteinases (MMPs) and heat shock proteins (HSPs) on the surface of blood plasma CD9-positive and FABP4-positive small extracellular vesicles (sEVs) from CRCPs depending on metabolic status, tumor angiogenesis, as to evaluate the sEVs markers as predictors of the effectiveness of thermoradiotherapy.

I suggest to the authors that in figures 1, 2, 3 and 4, the background of the “x” and “y” axes be thickened, for a clearer visibility.

In general, the quality of the article is good and, overall, the manuscript is interesting to readers. English language and style are good, but there are some minor spelling mistakes. In conclusion, I consider the article could be a useful contribution to the journal. I recommend the manuscript for being published.

Reviewer 2 Report

This work provides new information about how the composition of small extracellular vesicles (sEVs) in the blood plasma of colorectal cancer patients reflects the presence of the metabolic syndrome and correlates with angiogenesis and the effectiveness of thermoradiation therapy. The FABP4+MMP9+MMP2-TIMP1- population of plasma sEVs would be useful in monitoring patients after treatment for early detection of tumor progression, which would be accompanied by increased neoangiogenesis.

Reviewer 3 Report

The authors analyzed expression of MMPs and HSPs in CD9/Fabp4 positive EV's. The study focuses on analysis of samples isolated from patients with variable cancer stage and metabolic status to predict outcomes of thermoradiation therapy and tumor angiogenesis. The study is well written, methods are described in great detail. The study will be of interest to a general audience due to its translational potential.
